# Chemoselective single-site Earth-abundant metal catalysts at metal–organic framework nodes

Kuntal Manna[1],*, Pengfei Ji[1],*, Zekai Lin[1], Francis X. Greene[1], Ania Urban[1], Nathan C. Thacker[1] & Wenbin Lin[1]

Earth-abundant metal catalysts are critically needed for sustainable chemical synthesis. Here we report a simple, cheap and effective strategy of producing novel earth-abundant metal catalysts at metal–organic framework (MOF) nodes for broad-scope organic transformations. The straightforward metalation of MOF secondary building units (SBUs) with cobalt and iron salts affords highly active and reusable single-site solid catalysts for a range of organic reactions, including chemoselective borylation, silylation and amination of benzylic C–H bonds, as well as hydrogenation and hydroboration of alkenes and ketones. Our structural, spectroscopic and kinetic studies suggest that chemoselective organic transformations occur on site-isolated, electron-deficient and coordinatively unsaturated metal centres at the SBUs via σ-bond metathesis pathways and as a result of the steric environment around the catalytic site. MOFs thus provide a novel platform for the development of highly active and affordable base metal catalysts for the sustainable synthesis of fine chemicals.

[1] Department of Chemistry, The University of Chicago, 929 E 57th Street, Chicago, Illinois 60637, USA. * These authors contributed equally to this work. Correspondence and requests for materials should be addressed to W.L. (email: wenbinlin@uchicago.edu).

Metal oxides, metal alloys and metal nanoparticles dispersed on oxide supports form the cornerstone of heterogeneous catalysts used in chemical and petrochemical industries[1,2]. However, the presence of multiple active sites makes selectivity difficult to achieve in traditional heterogeneous catalysis. Significant effort has thus been devoted to developing single-site heterogeneous catalysts[3,4]. The simplest and most common method of producing single-site heterogeneous catalysts is to anchor catalytically active atoms, ions or molecular complexes to high surface area solid supports, such as silica or other robust metal oxides. However, this approach often leads to non-uniform distribution of catalysts throughout the solid supports, which can be complicated by intermolecular interactions between catalytic sites and the interactions of active sites with surface functionalities. Preventing intermolecular deactivation has been a challenge in the development of novel base metal catalysts, since homogeneous base metal catalysts generally need to be stabilized with bulky, elaborately designed ligands that can impede their catalytic activity. In this work, we attempt to use inorganic oxide clusters that are well positioned and separated in porous metal–organic frameworks (MOFs) to design highly active earth-abundant metal-based single-site solid catalysts.

Built out of metal cluster secondary building units (SBUs) and organic linkers, MOFs have been explored as useful molecular materials for many potential applications, including gas storage[5,6], separation[7,8], catalysis[9,10], sensing[11–13], biomedical imaging[14], drug delivery[15], solar energy harvesting[16] and conductivity[17,18]. By utilizing pre-metalated organic struts or via postsynthetic metalation of the functionalized bridging linkers, MOFs provide a highly tunable platform to engineer single-site solid catalysts for many organic transformations that cannot be performed by traditional porous inorganic materials[19–23]. The diversity of metal cluster SBUs offers an alternative strategy for generating MOF catalysts. Although MOF SBUs have been used as acid catalysts[24,25], their application in more challenging catalytic reactions or as potential supports for catalysis has been far less explored[26,27].

Herein, we report a simple strategy for discovering highly active base metal catalysts functionalized at metal–organic framework nodes (M@SBU). The straightforward and cost-effective metalation of SBUs of UiO-MOFs with readily available cobalt and iron precursors affords highly electron-deficient and coordinatively unsaturated metal centres that can catalyse interesting organic reactions. We believe this occurs via σ-bond metathesis reaction pathways. The Co@SBU and Fe@SBU MOF materials are highly active and reusable single-site solid catalysts for site-selective $sp^3$ C–H functionalization reactions, such as undirected benzylic C–H borylation, silylation and amination. In addition, we have also discovered the first example of catalytic undirected $sp^3$ C–H silylation with alkoxysilane using the Co@SBU catalyst.

## Results

### Synthesis and characterization of M@UiO Materials.
UiO-68 was synthesized via a solvothermal reaction between $ZrCl_4$ and triphenyldicarboxylic acid ($H_2TPDC$) in the presence of DMF ($N,N$-dimethylformamide) and trifluoroacetic acid in 95% yield (Supplementary Figs 1–3)[28]. The deprotonation of $Zr_3(\mu_3\text{-OH})$ sites in SBUs of UiO-68 with $n$BuLi followed by reaction with $CoCl_2$ or $FeBr_2 \cdot 2THF$ in THF afforded the Co- or Fe-functionalized UiO materials (UiO-CoCl and UiO-FeBr) as a deep blue or brown solid, respectively (Fig. 1a and Supplementary Figs 4–15). Crystallinity of UiO-68 was maintained upon metalation, as suggested by similarities in the powder X-ray

diffraction patterns of UiO-68, UiO-CoCl and UiO-FeBr (Fig. 1b,e). Inductively coupled plasma-mass spectrometry (ICP-MS) analysis of the digested UiO-CoCl and UiO-FeBr revealed 100% metalation at the $Zr_3(\mu_3\text{-OH})$ sites, corresponding to four Co/Fe centres per $Zr_6$ node. Infrared spectrum of UiO-CoCl showed the disappearance of $v_{\mu 3O-H}$ band ($\sim 3,640$ cm$^{-1}$, KBr), consistent with metalation of $Zr_3(\mu_3\text{-OH})$ sites (Supplementary Fig. 9). In addition, the TEM-EDX analysis indicated that cobalt and iron are uniformly distributed throughout the particles (Fig. 1d, Supplementary Figs 10 and 12). UiO-66 and UiO-67 were functionalized with $CoCl_2$ in a similar fashion as UiO-68 (Supplementary Figs 16–17).

A single-crystal X-ray diffraction study revealed that UiO-CoCl crystallizes in the $Fm\bar{3}m$ space group, with the $Zr_6$ $(\mu_3\text{-O})_4(\mu_3\text{-OH})_4$ SBUs connected by the TPDC bridging linkers to afford the 12-connected fcu topology. The void space was calculated to be 79.6% by PLATON (Supplementary Fig. 19 and Supplementary Table 1). However, due to the crystallographic disorder of the CoCl moiety, the Co coordination environments in UiO-CoCl could not be established by X-ray crystallography (Supplementary Fig. 20). Instead, we used X-ray absorption spectroscopy to investigate the coordination environments of Co and Fe. The oxidation states of UiO-CoCl and UiO-FeBr are +2, as determined by comparing the energies of the pre-edge peaks to the Co(II) and Fe(II) reference compounds (Supplementary Figs 21–29). Fitting the extended X-ray absorption fine structure (EXAFS) regions of UiO-CoCl and UiO-FeBr confirmed that the Co and Fe centres were coordinated to three SBU oxygen atoms (one oxide moiety that is triply bridged to three Zr centres and two TPDC carboxylate oxygen atoms that are doubly bridge the Zr centres) and one halogen atom as shown in the model complexes (Fig. 1c,f; Supplementary Tables 2–4).

### UiO-Co-catalysed undirected benzylic C–H borylation.
Upon treatment with $NaEt_3BH$, UiO-Co became an active catalyst in undirected dehydrogenative borylation of benzylic C–H bonds using $B_2(pin)_2$ (pin = pinacolate) or HBpin as the borylating agents. Borylation of alkyl C–H bonds provides alkyl boronates, which are versatile reagents in organic synthesis[29,30]. Although selective borylation of benzylic C–H bonds has been achieved with a few homogeneous catalysts based on precious metals such as Pd[31], Rh[32] or Ir[33], the earth-abundant metal-catalysed undirected and chemoselective benzylic C–H borylation is rare[34]. Recently, α-di-imine cobalt catalysts were reported for benzylic C–H borylation at high catalyst loadings (5–30 mol%) and with long reaction times[35]. The UiO-Co catalysed borylation reactions were first screened by varying temperatures and solvents, which revealed high catalytic activities and selectivities when the borylation reactions were performed using $B_2(pin)_2$ in neat alkylarenes at 103 °C (Supplementary Tables 5 and 6). Under optimized reaction conditions, primary benzylic boronate esters were afforded in excellent yields from a range of methylarenes with UiO-Co (0.2 mol% Co) (Table 1). The borylation reactions of $m$-xylene and toluene afforded corresponding boronate esters in good yields with excellent selectivities for the benzylic boronates over the aryl boronates (entries 1–2, Table 1). The powder X-ray diffraction patterns of UiO-Co recovered from the borylation reaction of $m$-xylene remained the same as that of freshly prepared UiO-Co (Fig. 1b), indicating that the MOF frameworks are stable under the catalytic conditions. Only benzylic boronates were obtained in good yields from mesitylene, $p$-xylene and 4-tert-butyl-toluene, presumably because the steric hindrance at the aryl C–H bonds was greater than that of benzylic positions (entries 3–9, Table 1). TONs as high as 2,300 were obtained for the borylation of mesitylene.

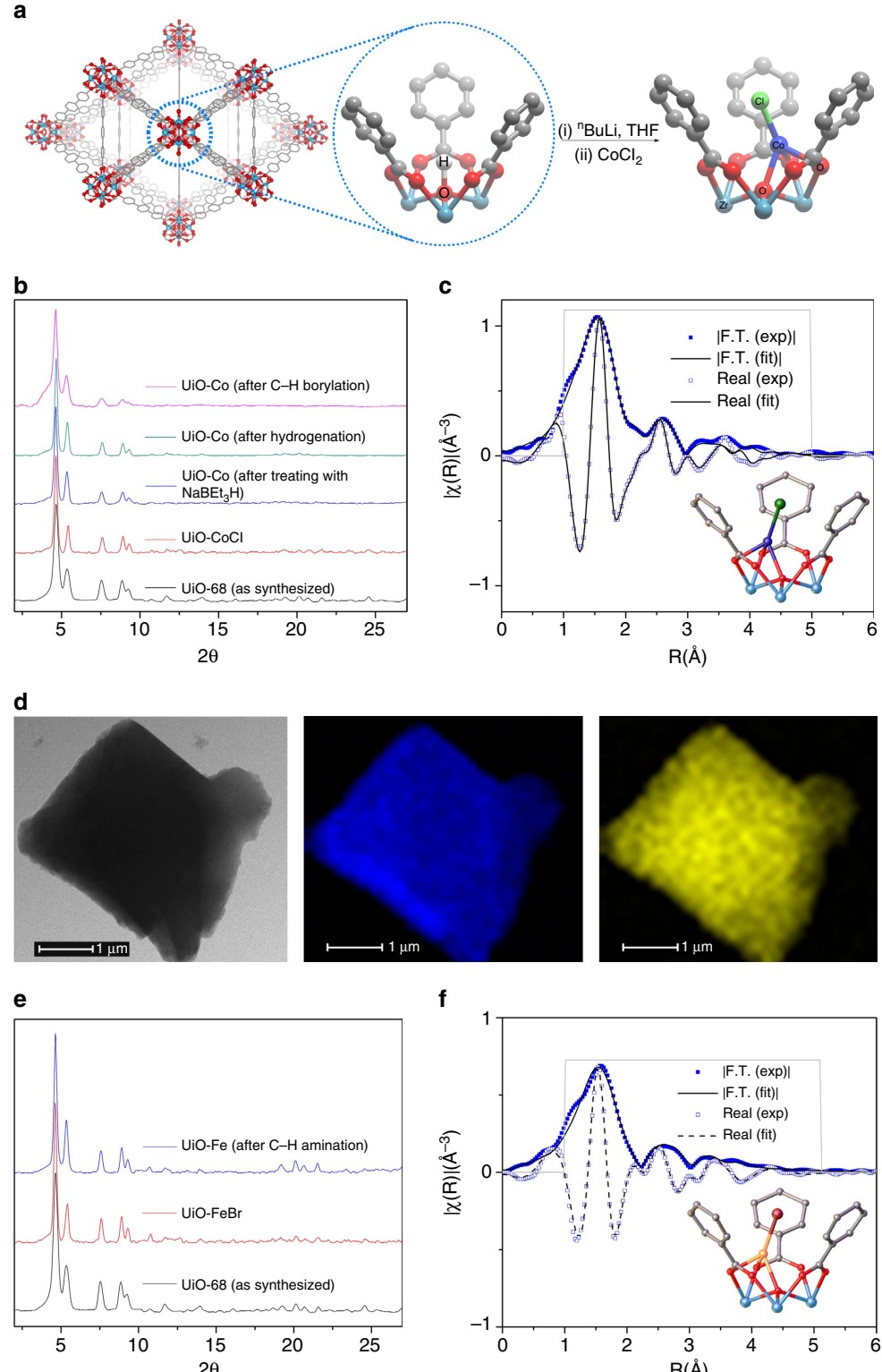

**Figure 1 | Characterization of Co- and Fe-functionalized UiO-68.** (**a**) Scheme showing the postsynthetic metalation of the SBUs of UiO-68 with CoCl$_2$. (**b**) The similarities between the powder X-ray diffraction patterns of UiO-68 (black), UiO-CoCl (red), UiO-Co (blue) and UiO-Co recovered from hydrogenation of 1-octene (green) and recovered from C–H borylation of m-xylene (pink) indicate the retention of UiO-68 crystallinity after postsynthetic metalation and catalysis. (**c**) EXAFS spectra and the fits in R-space at the Co K-edge of UiO-CoCl showing the magnitude (solid squares, solid line) and real component (hollow squares, dashed line) of the Fourier transform. The fitting range is 1–5.0 Å in R space (within the grey solid lines). (**d**) TEM-EDX mapping of UiO-Co: TEM image (left), Co distribution is shown in blue (middle), and Zr distribution is shown in yellow (right). TEM-EDX mapping of UiO-Co indicates that Co and Zr are evenly distributed throughout the MOF-particle. (**e**) The similarity of the powder X-ray diffraction patterns of UiO-68 (black), UiO-FeBr (red), and UiO-Fe recovered from C–H amination of tetralin (blue) indicates the retention of UiO-68 crystallinity after postsynthetic metalation and catalysis. (**f**) EXAFS spectra and the fits in R-space at the Fe K-edge of UiO-FeBr showing the magnitude (solid squares, solid line) and real component (hollow squares, dashed line) of the Fourier transform. The fitting range is 1–5.1 Å in R space (within the grey solid lines).

**Table 1 | UiO-Co-catalysed benzylic C–H borylation.***

| Entry | Substrate | Product(s) | % Co-loading | Time | % Yield† (Benzyl:Ar) |
|---|---|---|---|---|---|
| 1 | | | 0.2 | 2.5 d | 92 (96:4) |
| 2 | | | 0.2 | 5 d | 72 |
| 3 | | | 5 | 6 h | 92 |
| 4 | | | 0.2 | 2 d | 100 |
| 5 | | | 0.05 | 5 d | 94 |
| 6 | | | 0.025 | 12 d | 58 |
| 7 | | | 5 | 7 h | 93 |
| 8 | | | 0.2 | 2 d | 96 |
| 9 | | | 0.2 | 6 d | 76‡ |
| 10 | | | 0.2 | 2.5 d | 83 (84:12) |
| 11 | | | 0.2 | 5 d | 36 |
| 12 | | | 0.2 | 3 d | 70‡ (78:22) |
| 13 | | | 0.2 | 6 d | 81‡ (80:20) |

*Reaction conditions: 1.0 mg of UiO-68-CoCl, 5 equivalent NaBEt$_3$H (1.0 M in THF) w.r.t. Co, arene (2 mL, neat), B$_2$pin$_2$, 103 °C, N$_2$.
†Isolated yield.
‡The yield was determined by $^1$H NMR after oxidizing the boronate ester to the corresponding alcohol.

However, reactions with electron-deficient methylarenes such as *m*-chlorobenzene were slow. Notably, UiO-Co-catalysed borylation occurred not only at primary benzylic C − H bonds, but also at secondary and tertiary benzylic C − H bonds. At 0.2 mol% Co loading, UiO-Co afforded secondary and tertiary benzylic boronate esters from ethylbenzene and isopropryl benzene, respectively, in 100% conversion, and the selectivities for the benzylic products over the aryl products were ~4:1 (entries 12–13, Table 1). These results contrast with analogous reactions reported with a chemoselective homogeneous iridium catalyst that is active for borylation of only primary benzylic C − H bonds[33].

Unlike homogeneous catalysts, which cannot be easily reused, we found that, at a 1.0 mol% Co loading, the UiO-Co catalyst could be used at least five times in the borylation of *p*-xylene (Supplementary Fig. 30). Furthermore, the boronate ester was obtained in high purity simply by removing the solid catalyst and the organic volatiles. The heterogeneity of UiO-Co was confirmed by several experiments. The leaching of Co and Zr into the supernatant was very low—0.14 and 0.056%, respectively—during the course of the borylation reaction, as shown by ICP-MS analysis. Moreover, no further conversion was detected after removal of UiO-Co from the reaction mixture (Supplementary Discussion). Our control experiment ruled out the contribution of catalytic activity of any trapped Co nanoparticles within the MOFs (Supplementary Discussion). In addition, the rate of the borylation reaction of *p*-xylene was significantly higher than that of analogous bulkier alkenes 4-*tert*-butyl-toluene and 3,5-di-*tert*-butyl-toluene under identical conditions, which demonstrates that catalysis is facilitated by Co@SBU sites both inside the pores and on the surface of the MOFs (Supplementary Table 8 and Supplementary Discussion).

Having shown the highly selective benzylic C − H borylation of alkylarenes with the UiO-Co catalyst, we sought to investigate the nature of the catalytic species and to reveal their mechanism using structural, kinetic and spectroscopic techniques. The treatment of UiO-CoCl with NaEt$_3$BH in THF likely affords the 'Zr$_3$-($\mu_4$-O)-Co-H' (UiO-CoH) species. The reaction of UiO-CoH with HBpin readily generates UiO-Co-Bpin and an equivalent amount of H$_2$ (Supplementary Fig. 32). UiO-Co-Bpin can also be

prepared by the reaction of UiO-CoCl with $B_2pin_2$ at 100 °C for 12 h. X-ray absorption near edge structure analysis indicated that the Co centres in both UiO-CoH and UiO-Co-Bpin are in $+2$ oxidation states (Supplementary Fig. 21). EXAFS fitting of UiO-CoH or UiO-Co(Bpin) revealed a Co local coordination environment similar to UiO-CoCl, where Co is coordinated to

three SBU oxygen atoms and one hydride or one Bpin, respectively (Fig. 2a and Supplementary Figs 25–27).

We also prepared the Co-functionalized oxozirconium methacrylate cluster $Zr_6(OCoCl)_4O_4(OMc)_{12}$ (Mc, methacrylate, Fig. 2b, Supplementary Fig. 18 and Supplementary Methods)[36] as a homogeneous control to test whether Co@SBU is a

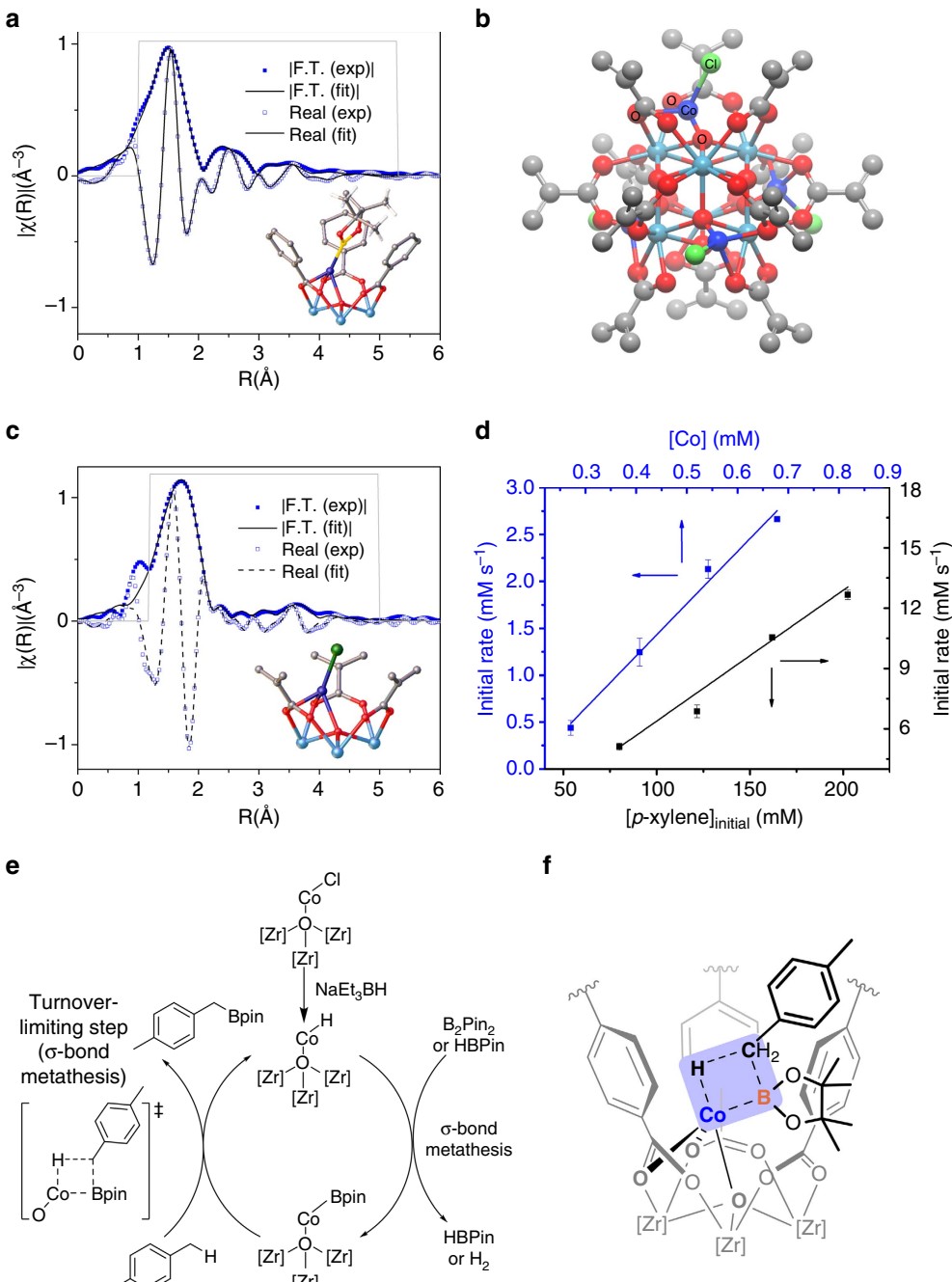

**Figure 2 | Mechanistic study of Co@SBU catalysed reactions.** (**a**) EXAFS spectra and the fits in R-space at the Co K-edge of UiO-68-CoBpin. Solid squares and solid line show the magnitude of of the Fourier transform whereas hollow squares and dashed line correspond to the real component (hollow squares, dashed line) of the Fourier transform. The fitting range is 1–5.3 Å in R space (within the grey solid lines). (**b**) Structural model of homogeneous $Zr_6O_4(OCoCl)_4(McO)_{12}$ cluster. (**c**) EXAFS spectra and the fits in R-space at the Co K-edge of $Zr_6O_4(OCoCl)_4(McO)_{12}$ cluster. Solid squares and solid line show the magnitude of the Fourier transform whereas and real component (hollow squares and dashed line corespond to the real component of the Fourier transform. The fitting range of 1.18–4.98 Å in R space (within the grey solid lines) is shown. (**d**) Kinetic plots of initial rates-(d[$p$-xylene]/d$t$) for benzylic C − H borylation of $p$-xylene versus catalyst concentration and [$p$-xylene]$_{initial}$ for the first 12 h, showing the first-order dependence on both components. (**e**) According to the kinetic and experimental studies, UiO–Co-catalysed benzylic C − H borylation of $p$-xylene likely proceeds via [$2\sigma + 2\sigma$] cycloaddition of 'Co-Bpin' bond with 'H–CH$_2$Ph' bond of $p$-xylene as the turnover-limiting step. (**f**) Structural model showing the likely origin of chemoselectivity for C–H borylation reactions in the turnover-limiting step.

competent catalyst for the borylation reactions. EXAFS analysis indicated that the Co coordination environment of $Zr_6(OCoCl)_4O_4(OMc)_{12}$ is similar to that of UiO-CoCl (Fig. 2c, Supplementary Fig. 28, and Supplementary Table 3). Interestingly, on activation with $NaEt_3BH$, this homogeneous $Zr_6$-based Co-cluster was also active in catalysing C−H borylation of p- and m-xylenes with selectivities for the benzylic boronates over the aryl boronates (Supplementary Table 7), similar to those observed for UiO-Co, suggesting that the SBUs were the sites of catalysis. However, the reaction rate and the chemoselectivity in the borylation of m-xylene were lower for the homogeneous catalyst than those of UiO-Co. Furthermore, UiO-Co was more active and chemoselective than analogous UiO-MOFs with smaller pore sizes, such as UiO-67-Co and UiO-66-Co (entries 1–3, Supplementary Table 5). These results demonstrate that the porous frameworks and the pore sizes of UiO-Co are crucial in controlling the chemoselectivity and the rate of the reaction.

To further explore the mechanism, the empirical rate law was determined by the method of initial rates (<10% conversion), which showed that the C−H borylation of p-xylene catalysed by UiO-Co has a first-order dependence on the catalyst and p-xylene

concentrations (Fig. 2d) and a zeroth-order dependence on the $B_2pin_2$ concentration (Supplementary Fig. 33). In addition, the conversion of the deuterated p-xylene was slower than the proteo-p-xylene. Primary kinetic isotope effects from initial substrate conversion measurements $[k'^{(H)}_{obs}/k'^{(D)}_{obs} = 1.73(9)]$ indicated that the C−H bond cleavage of p-xylene is likely the turnover-limiting step (Supplementary Fig. 34). Furthermore, we did not observe any deuterium incorporation into the benzyl boronate, which suggests that the C–H bond activation is irreversible. The oxidative addition of C–H bond to the Co-centre is unlikely to occur in the turnover-limiting step because the $Co^{2+}$ centre was electron poor and the borylation of the benzylic C−H bonds was faster than that of the more electron-rich aryl C−H bonds of alkylarenes. Instead, UiO-Co-catalysed benzylic borylation likely proceeds via σ-bond metathesis pathways. On the basis of our experimental observations, we propose the following mechanism (Fig. 2e): The reaction of '$Zr_3$-$\mu_4$-O-Co-Cl' moiety in UiO-CoCl with $NaEt_3BH$ generates the active species UiO-CoH in the cycle. The σ-bond metathesis between Co-H and $B_2pin_2$ produces $Zr_3$-$\mu_4$-O-Co-Bpin species, which then reacts with the alkylarene via a four-centred turnover-limiting step involving $[2\sigma + 2\sigma]$ cycloaddition of a

---

**Table 2 | UiO-Co-catalysed undirected benzylic C–H silylation.***

| Entry | Product(s) | % Co-loading | Silane | Time | Yield (Benzyl:Ar) |
|---|---|---|---|---|---|
| 1 | | 0.2 | $Et_3SiH$ | 2.5 d | 74 (60:40)[†] |
| 2 | | | $(OEt)_3SiH$ | 3 d | 81[†] |
| 3 | | 0.4 | $(OEt)_3SiH$ | 3 d | 89 |
| 4 | | 0.2 | $Et_3SiH$ | 2 d | 66[†] |
| 5 | | | $(OEt)_3SiH$ | 3 d | 71[†] |
| 6 | | 0.2 | $(OEt)_3SiH$ | 4 d | 80[†] |
| 7 | | 0.2 | $Et_3SiH$ | 2 d | 62 (96:4)[†] |
| 8 | | 0.2 | $(OEt)_3SiH$ | 6 d | 84[†] |
| 9 | | 0.2 | $Et_3SiH$ | 4 d | 46[†] |
| 10 | | 0.2 | $(OEt)_3SiH$ | 3 d | 52 |
| 11 | | 0.2 | $(OEt)_3SiH$ | 3 d | 61[†] |
| 12 | | 0.2 | $Et_3SiH$ | 3 d | 77[†](87:13)[‡] |

*Reaction conditions: 1.0 mg of UiO-68-CoCl, 5 equivalent $NaBEt_3H$ (1.0 M in THF) w.r.t. Co, arene (2 ml, neat), silane, 98 °C, heated under $N_2$.
[†]Isolated yield was determined after oxidizing the silane product to corresponding alcohol via Tamao-Fleming oxidation.
[‡]Ratio of primary to tertiary alcohol.

'Co–Bpin' bond with the 'H–CH$_2$Ar' bond of alkylarene to furnish the benzyl boronates and regenerate UiO-CoH. Although the electron-deficient Co catalyst should favour the electron-rich aryl C-H bonds, the steric hindrance from the three phenyl rings surrounding the Co centre directs selective binding of less-hindered benzylic C–H bonds (Fig. 2f and Supplementary Fig. 31).

**UiO-Co-catalysed undirected benzylic C − H silylation.** Inspired by the discovery of UiO-Co catalysed selective borylation of the benzylic C − H bonds, we attempted to develop undirected benzylic C − H silylation reactions using alkyl- and alkoxysilanes. Although significant progress has been made in developing catalysts for aryl C − H silylation, the silylation of alkyl C − H bonds has been far less explored[37,38]. In particular, examples of undirected intermolecular alkyl C − H silylation are few[39–41], and there is no report of such transformation with trialkoxysilanes. Direct installation of a trialkoxysilyl group via alkyl C − H bond activation generates alkyltrialkoxysilanes, which are not only useful precursors to commercial polymers but also widely used as silylating agents for surface functionalization and as nontoxic transmetalation agents in cross-coupling reactions. We used UiO-Co as the active catalyst in the silylation of benzylic C − H bonds with Et$_3$SiH or (OEt)$_3$SiH (Table 2 and Supplementary Figs 35–38). Heating the mixture of UiO-Co (0.2–0.4 mol% Co) and silane in neat alkylarene at 98 °C gave the corresponding alkylsilanes. Owing to the difficulty of isolating some silylated products, the crude products were directly oxidized

via Tamao–Fleming oxidation, and the final products were isolated as the corresponding benzyl alcohols[42,43]. In the case of silylation of *m*-xylene and ethylbenzene with Et$_3$SiH, we observed good selectivities of silylation in favour of benzylic C − H bonds over aryl C − H bonds, resulting in high yields of benzyl alcohols. Interestingly, reactions of (OEt)$_3$SiH occurred exclusively with benzylic C − H bonds of the alkylarenes, as shown by GC-MS and $^1$H NMR spectroscopy, providing corresponding benzylic alcohols after oxidation in 62–89% yield.

**UiO-Co-catalysed hydrogenation and hydroboration reactions.** UiO-Co is also highly active for catalytic hydrogenation of a range of olefins at room temperature (Table 3). Mono-substituted alkenes such as 1-octene and styrene were readily hydrogenated in quantitative yields with TONs > $1.0 \times 10^5$ (entries 1–4, Table 3). At 0.1–0.01 mol% Co-loading, UiO-Co catalysed hydrogenation of 1,1-, *cis*-1,2-disubstituted alkenes, α-iso-propylstyrene and cyclohexene in quantitative yields (entries 5–9, Table 3 and Supplementary Fig. 39). In addition, dialkenes (allyl ether), tri-substituted alkenes (*trans*-α-methylstilbene) and carbonyl-functionalized alkenes (dimethyl itaconate) were quantitatively hydrogenated in excellent yields (entries 10–13, Table 3). Within only 66 h, UiO-Co displayed an exceptional TON of $3.54 \times 10^6$ in hydrogenation of 1-octene, the highest TON ever reported for an earth-abundant metal-catalysed olefin hydrogenation (entry 2, Table 3). Furthermore, only 3.7 p.p.m. Co and 1.7 p.p.m. Zr remained in the *n*-octane product after simple filtration. UiO-Co showed impressive recyclability: it

**Table 3 | UiO-Co-catalysed hydrogenation of olefins.\***

| Entry | Substrate | % Co-loading | Time | Yield (%)$^†$ | TONs |
|---|---|---|---|---|---|
| 1 | | 0.002 | 0.5 h | 100 | >5.0×10$^4$ |
| 2 | | 2.65 ×10$^{-5}$ | 66 h | 94 | 3.54×10$^6$ |
| 3 | Ph | 0.01 | 0.5 h | 100 | >1.0×10$^4$ |
| 4 | | 0.001 | 7 h | 100 (96) | >1.0×10$^5$ |
| 5 | Ph / Me | 0.1 | 1 h | 100 | >1.0×10$^3$ |
| 6 | | 0.01 | 14 h | 100 (90) | >1.0×10$^4$ |
| 7 | Ph | 0.1 | 15 h | 100 | >1.0×10$^3$ |
| 8 | Ph | 0.1 | 12 h | 100 | >1.0×10$^3$ |
| 9 | | 0.1 | 30 h | 100 | >1.0×10$^3$ |
| 10 | Ph / Ph | 0.1 | 48 h | 100 (100) | >1.0×10$^3$ |
| 11$^‡$ | | 0.002 | 72 h | 76 | 3.8×10$^4$ |
| 12 | O | 0.1 | 72 h | 93 (76) | 930 |
| 13 | MeO$_2$C / CO$_2$Me | 0.5 | 72 h | 91 | 182 |

\*Reaction conditions: 1.0 mg of UiO-CoCl, 5 equivalent of NaBEt$_3$H (1.0 M in THF) w.r.t. Co, alkene, THF, 40 bar H$_2$, 23 °C.
$^†$The yields were determined by $^1$H NMR with mesitylene as the internal standard. Isolated yield given in the parenthesis.
$^‡$Reaction was performed at 60 °C.

could be recovered and reused without any loss of catalytic activity at least 16 times for the hydrogenation of 1-octene at a 0.01 mol% Co loading (Supplementary Fig. 40). The powder X-ray diffraction patterns of UiO-Co after hydrogenation were the same as those of the pristine MOF catalysts, which verified the stability of the framework under catalytic conditions (Fig. 2b). Hardly any metal was leached after the first run, according to ICP-MS analyses of the organic product, with leachings of only 0.9% for Co and 1.0% for Zr. Furthermore, no conversion of alkene was observed after the removal of the solid catalyst, demonstrating that the leached Co was not responsible for the catalytic activity (Supplementary Discussion).

We also evaluated the UiO-Co for catalytic hydroboration of alkenes and carbonyl compounds (Table 4 and Supplementary Figs 41–42). We examined hydroboration reactions by treating alkenes, ketones or aldehydes with HBpin and 0.01–0.4 mol% UiO-Co at 60–100 °C. UiO-Co resulted in borate ester products from several carbonyl substrates, including alkyl and alkoxy-functionalized aryl ketones and aldehydes in 81–98% yields with TONs up to $5.4 \times 10^4$ (Table 4). Alkenes such as 1-octene, styrene and α-methylstyrene were hydroborated selectively with 0.1–0.4 mol% UiO-Co in anti-Markovnikov manner to give corresponding alkylboronates in extremely high yields (Table 4). We obtained pure hydroboration products through a simple process of removing the catalyst by centrifuge, then removing the organic volatiles.

**UiO-FeBr catalysed $sp^3$ C–H amination.** We next examined the use of UiO-68 in other base metal-catalysed organic transformations. We studied the catalytic activity of UiO-FeBr in C–H amination reactions because the conversion of C–H bonds to C–N bonds provides a valuable method for introducing nitrogen functionalities directly into a molecule[44–47]. Developing iron catalysts for $sp^3$ C–H amination is of particular interest due to the high abundance and low toxicity of iron[48–50]. Although most Fe-catalysed C–H amination reactions use sulfamides or azides as a nitrogen source[45,46,48,50,51], we report here the first example of using aniline as a nitrogen source. Heating aniline with neat substrates such as indane, tetraline and cyclohexene gave a C–H aminated product in 41–53% yield with UiO-FeBr

(2–10 mol% Fe, Table 4 and Supplementary Fig. 43). We propose a catalytic process involving a one-electron Fe(II)–Fe(III) cycle, which is analogous to the well-studied copper-catalysed C–H amination reactions that are believed to proceed through a Cu(I)–Cu(II) cycle (Supplementary Fig. 44)[47]. The inverse dependence of conversions on the substrate sizes (Table 5, entries 1–3) is consistent with the notion that the $sp^3$ C–H amination occurs at the UiO-FeBr site inside the MOF owing to the sluggish diffusion of large substrates through MOF channels.

## Discussion

The deprotonation of $\mu_3$–OH sites of MOFs SBUs followed by reactions with iron- and cobalt-halides afforded Fe- and Co-functionalized MOF-materials for a broad scope of organic transformations (Fig. 3). The treatment of UiO-CoCl with NaEt$_3$BH generated highly robust $\mu_4$–O–Co(H) species at MOF nodes due to the prevention of intermolecular decomposition pathways. All the functionalized MOF materials were characterized by powder X-ray diffraction, ICP-MS, TEM-EDX, X-ray absorption spectroscopy, TGA and BET analysis. The UiO carboxylate groups did not react with $n$BuLi during lithiation and metalation as evident by the observation of only H$_2$TPDC in the $^1$H NMR spectrum of the digested metalated UiO-68 (Supplementary Fig. 8 and Supplementary Discussion) and by the retention of strong carboxylate carbonyl stretching peaks at 1,605 and 1,591 cm$^{-1}$ in the infrared (KBr) spectrum (Supplementary Fig. 9a). EXAFS of UiO-CoCl and UiO-FeBr suggested that the Co(II) and Fe(II) centres in the MOFs were coordinated to three SBU oxygen atoms: one oxide moiety that is triply bridged to three Zr centres and two TPDC carboxylate oxygen atoms that doubly bridge the Zr centres and one halogen atom. The very-weak-field coordination environment consisting of O-atoms that the UiO-M (M = Co, Fe) framework provides is very different than those accessible to molecular catalysts that typically consist of stronger field N- or P-donor atoms. Indeed, bipyridine and phenanthroline-based MOF-Co catalysts were recently reported to be active in borylation of arene C–H bonds instead of benzylic C–H bonds[52]. Beyond the low coordination number, weak-field catalyst sites that lead to high electrophilicity

---

**Table 4 | UiO-Co-catalysed hydroboration of aldehydes, ketones and alkenes.\***

*Reaction conditions: UiO-CoCl, 5 equiv of NaBEt$_3$H (1.0 M in THF) w.r.t. Co, substrate, HBpin (1.4 equiv w.r.t. substrate), 60 – 100 °C. Yields were isolated yields; TON calculated based on $^1$H NMR yield.

---

**Table 5 | UiO-FeBr-catalysed $sp^3$ C–H amination.\***

| Entry | R-H | Product | Fe loading (%) | Yield (%)† |
|---|---|---|---|---|
| 1 | indane | HN–Ph aminoindane | 10 | 53 (45) |
| 2 | tetraline | HN–Ph aminotetraline | 10 | 49 |
| 3 | cyclohexene | N(H)–Ph aminocyclohexene | 2 | 41 |

*Reaction conditions: 2-10 mol% of UiO-FeBr,, C-H substrate (3.2 mmol, neat), aniline (0.16 mmol), ('BuO)$_2$ (0.48 mmol, 3.0 eq), 100 °C.
†Yields were determined by $^1$H NMR using MeNO$_2$ as internal standard. Isolated yield in the parenthesis.

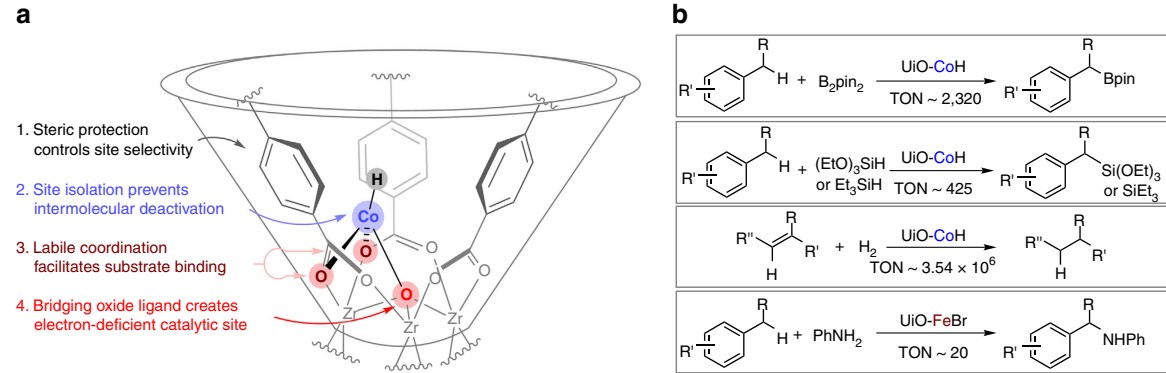

**Figure 3 | Conceptual approach to designing Co@SBU single-site solid catalysts for chemoselective $sp^3$ C–H functionalization and other reactions.** (**a**) An electron-deficient Co(II)-centre located in a unique sterically protected environment at the SBU of UiO-68 favours the activation of less-hindered alkyl C–H bonds over aryl C–H bonds via proposed σ-bond metathesis pathways. (**b**) The $sp^3$ C–H functionalization reactions and alkene hydrogenation catalysed by M@SBU catalyst (M = Co or Fe).

are possible in these MOFs but are simply not available to small molecule homogeneous catalyst systems because ligands based on weakly binding O-donors would readily dissociate from first row transition metals.

Our spectroscopic and kinetic studies suggest that the SBU-supported Co–H species can chemoselectively catalyse $sp^3$ C–H functionalization reactions via σ-bond metathesis pathways and as a result of the steric environment around the catalytic site. However, the nature of the active species has not been unambiguously established and will be the subject of future studies. The role of the MOF-framework and the strategy of using a specific pore size are also very important for chemoselective reactions. As discussed previously, UiO-Co was more active and chemoselective than homogeneous cluster control, $Zr_6(OCoCl)_4O_4(OMc)_{12}$ (Supplementary Tables 9–12), and also analogous UiO-MOFs with smaller pore sizes, such as UiO-67-Co and UiO-66-Co. These results suggest that the chemoselective C–H activation reactions occurred within the pore and were dependent on the open channel sizes because of competition between the reactions occurring at SBUs within the pore and the non-selective background reactions occurring on the surface, as well as on the different diffusion rates of the organic substrates through the open channels of varied sizes.

In summary, we developed a simple strategy of treating metal–organic framework nodes with readily available and cheap earth-abundant metal precursors to afford highly active and selective single-site solid catalysts for a broad scope of organic transformations, including novel site-selective borylation and silylation of $sp^3$ C − H bonds. The unique coordination environment of Co/Fe centres of the secondary building units as well as the porous frameworks and pore structures of UiO-MOF play important roles in controlling the rate and chemoselectivity of these organic reactions. Owing to the high stability of UiO-MOFs and the ease of functionalizing SBUs with metal ions, we anticipate that MOFs may offer a versatile platform for discovering new catalytic transformations and developing earth-abundant metal and other metal catalysts for sustainable synthesis of fine and commodity chemicals. The cost effective nature of the present M@SBU approach has the potential to move MOF catalysts from novel discoveries to practical applications.

## Methods

**General methods.** All of the catalytic reactions were carried out under nitrogen in a standard inert atmosphere with Schlenk techniques or inside a nitrogen-filled glovebox. Detailed procedures for the syntheses of ligands and MOFs are reported in the Supplementary Methods.

**Synthesis of UiO-68 and UiO-CoCl.** $ZrCl_4$ (1.30 mg, 5.03 μmol) and 1,4-bis (4-carboxyphenyl)benzene (1.6 mg, 5.53 μmol) were dissolved in 0.8 ml of DMF in a 1 dram vial, and 15.4 μl of trifluoroacetic acid was then added. The vial was capped and heated at 120 °C for 3 days to afford UiO-68 as a white solid (2.0 mg, 95% yield). In the glovebox, UiO-68 (20.0 mg) in 3 ml THF was cooled to − 30 °C for 30 min and 33 μl of $n$BuLi (2.5 M in hexane) was added dropwise to the cold suspension. The resulting light yellow mixture was stirred slowly overnight, collected and washed with THF. Then the lithiated UiO-68 was transferred to a vial containing 5 ml THF solution of $CoCl_2$ (6.0 mg). The mixture was stirred overnight, collected and washed with THF to afford UiO-CoCl as a deep blue solid.

**Procedure for UiO-Co catalysed benzylic C–H borylation of methylarenes.** In a glovebox, UiO-CoCl (1.0 mg, 0.2 mol% Co) was charged into a small vial, to which 0.5 ml THF was added. Then, 15 μl NaBEt₃H (1.0 M in THF) was added to the vial, and the mixture was stirred slowly for 1 h in the glovebox. The solid was centrifuged out of suspension and washed twice with THF and then once with $p$-xylene. $B_2pin_2$ (43.0 mg, 0.169 mmol) in 2.0 ml $p$-xylene was added to the vial and the resulting mixture was transferred to a Schlenk tube. The tube was heated under nitrogen at 103 °C for 3 days. The reaction mixture was cooled to room temperature and the solid was centrifuged out of suspension. The extract was passed through a short plug of Celite and then concentrated *in vacuo* to give the pure boronate ester in 96% yield (75 mg, 0.324 mmol).

**Procedure for UiO-Co-catalysed benzylic C–H silylation.** In a glovebox, UiO-CoCl (1.0 mg, 0.2 mol% Co) was charged into a small vial, and 0.5 ml THF was added. Then, 15 μl NaBEt₃H (1.0 M in THF) was added to the vial, and the mixture was stirred slowly for 1 h in the glovebox. The solid was centrifuged out of suspension and washed twice with THF and then once with $p$-xylene. The solid suspended in 2 ml toluene was transferred to a Schlenk tube and (EtO)₃SiH (62.6 μl, 0.34 mmol) was added to the mixture. The tube was heated under nitrogen at 98 °C for 3 days. The reaction mixture was cooled to room temperature, and the solid was centrifuged out of suspension. The extract was concentrated *in vacuo* and then the residue was heated at 60 °C in vacuum for 3 h to give benzyltriethoxysilane as a colourless liquid in 89% yield (38 mg, 0.151 mmol).

**Procedure for UiO-Fe-catalysed C–H amination.** In a glovebox, UiO-FeBr (16 μmol Fe) was centrifuged off THF, washed twice with heptane and transferred into a Schlenk tube with indane (0.587 ml, 4.8 mmol). Aniline (0.0146 ml, 0.16 mmol) and di-*tert*-butylperoxide (0.0882 ml, 0.48 mmol) were directly added to the Schlenk tube, which was stirred at 100 °C for 3 days. The solid was then centrifuged out of suspension and washed twice with THF and the extract was concentrated under rotavap. The residue was purified by column chromatography on silica gel with 1% Et₃N and 5% EtOAc in Hexanes to afford the aminated product (16 mg, 0.076 mmol, 45%).

**Data availability.** The crystal structures reported are deposited at the Cambridge Crystallographic Data Centre (CCDC) under deposition numbers 1439497 and

1440158. The crystallographic files can be obtained free of charge from the Cambridge Crystallographic Data Centre via http://www.ccdc.cam.ac.uk/data_request/cif. All other data are available from the authors upon reasonable request.

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

## Acknowledgements

This work was supported by the NSF (CHE-1464941) and startup funds from the University of Chicago. We thank C. Poon and Dr A. Filatov for experimental help and Dr I. C. Hsiao for editing the manuscript. X-ray absorption spectroscopy analysis was performed at Beamline 9-BM, Advanced Photon Source (APS), Argonne National Laboratory (ANL). Use of the 9-BM beamline at the Advanced Photon Source, an Office of Science User Facility operated for the U.S. Department of Energy (DOE) Office of Science by Argonne National Laboratory, was supported by the U.S. DOE under Contract No. DE-AC02-06CH11357. Single-crystal diffraction studies were performed at ChemMatCARS, APS, ANL. ChemMatCARS is principally supported by the Divisions of

Chemistry (CHE) and Materials Research (DMR), NSF, under grant number NSF/CHE-1346572. Use of the Advanced Photon Source, an Office of Science User Facility operated for the U.S. DOE Office of Science by ANL, was supported by the U.S. DOE under Contract No. DE-AC02-06CH11357.

## Author contributions

K.M., P.J. and W.L. conceived this project, analysed the data and wrote the manuscript. K.M., P.J., Z.L., F.X.G., A.U. and N.C.T. performed the experiments and collated the data. All the authors discussed the results and commented on the manuscript. W.L. guided the whole project.

## Additional information

**Competing financial interests:** The authors declare no competing financial interests.

