## [Peer review file · Nature Communications]

Editorial Note: this manuscript has been previously reviewed at another journal that is not operating a transparent peer review scheme. This document only contains reviewer comments and rebuttal letters for versions considered at Nature Communications.
Following further discussion with Reviewer #2 after the second round of review, it was agreed to acknowledge the ambiguity over the exact nature of the active site.

Reviewers' comments:

Reviewer #1 (Remarks to the Author):

My original enthusiasm remains for this manuscript that was originally submitted to Nature Chemistry. The authors have addressed minor issues in the initial submission to Nature Chemistry. The manuscript is essentially ready for publication in Nature Communications.

The authors should explicitly mention their recent JACS article in which they have incorporated chelating N,N-donor ligands into a MOF framework for somewhat related Cu and Fe catalytic reaction. The present manuscript, however, stands on its own for the "minimal" ligand environment provided by the UiO-framework in the present work.

Additionally, the authors should explicitly mention a recent JACS article by the Chirik group that reports the use of molecular Co-based catalysts for C-H borylation.

Reviewer #2 (Remarks to the Author):

The authors have replied to the referee comments.

While the work is interesting and novel, substantial criticism was raised in terms of conceptual aspects and reliability of data. The long reaction times were seen as a major drawback and the benefit of MOF was questionable.

Despite the fact that the authors argue against this criticism their reply remains somewhat superficial. How can the similarity of PXRDs be a proof that BuLi does not react with a carboxylate? Certainly there are other more convincing experiments that can be carried out. It should be avoided to draw too many pictures that are not supported by hard facts. Amorphous side products are not detected in PXRDs and thus we only see an excerpt of reality.

Along the same line, statements arguing that BET surface areas are not important for catalysis are not very helpful and not convincing. Given the slow reaction rates, minority species in solution redepositing on amorphous supports could be the major active species without a hint for detectability.

In conclusion, the correspondence raises more questions and is not helpful to present a clear picture of achievements.

Author response to reviewer comments

Reviewer #1 (Remarks to the Author): My original enthusiasm remains for this manuscript that was originally submitted to Nature Chemistry. The authors have addressed minor issues in the initial submission to Nature Chemistry. The manuscript is essentially ready for publication in Nature Communications. The authors should explicitly mention their recent JACS article in which they have incorporated chelating N,N-donor ligands into a MOF framework for somewhat related Cu and Fe catalytic reaction. The present manuscript, however, stands on its own for the "minimal" ligand environment provided by the UiO framework in the present work. Additionally, the authors should explicitly mention a recent JACS article by the Chirik group that reports the use of molecular Co-based catalysts for C-H borylation.

Response: We thank the reviewer for the strong endorsement of our work. We have explicitly mentioned these two references in the manuscript (Ref 35 and 52).

For ref 35, we added "Recently, α -diimine cobalt catalysts were reported for benzylic C-H borylation although high catalyst loadings (5-30 mol%) and long reaction times." on Page 3.

For ref 52, we added "Indeed, bipyridine and phenanthroline-based MOF-Co catalysts were recently reported to be active in borylation of arene C-H bonds instead of benzylic C-H bonds." on Page 8.

Reviewer #2 (Remarks to the Author): The authors have replied to the referee comments. While the work is interesting and novel, substantial criticism was raised in terms of conceptual aspects and reliability of data. The long reaction times were seen as a major drawback and the benefit of MOF was questionable.

Response: We thank this reviewer for positive comments of our work and, more importantly, for the critique of our work. We have carried out several additional experiments to DIRECTLY address these critical comments; the results from these additional experiments have significantly strengthened the manuscript.

The UiO-Co catalysts are in fact very active. Long reaction times are only needed when we try to maximize TONs using extremely low catalyst loadings. The catalytic reactions are finished in very short periods of times when catalyst loadings corresponding to those used in state-of-the-art homogeneous catalysts are used. We carried out additional catalytic reactions of benzylic C-H borylation and alkene hydrogenation reactions at higher Co-loadings and in much shorter reaction times.

As shown in Table 1 (entries 3 and 7), at 5 mol% UiO-Co loading, benzylic C-H borylation reactions gave 92-93% yields in 6-7 h (TON ~20). These benzylic C-H borylation activities are much higher than those of the α -diimine cobalt catalysts (5-30 mol% catalysts and 2-5 days of reaction times in most cases to afford lower yields of products) that were published after the submission of this work to Nat. Chem. (Ref 35).

As shown in Table (entries 1, 3, and 5), even at 0.002-0.1 mol% UiO-Co loadings, hydrogenation of various alkenes was completed in 0.5-1.0 h. The UiO-Co catalysts are the most active earth-abundant hydrogenation catalysts reported to date.

In terms of benefits of MOF catalysts, we would like to summarize key aspects below:

1. The very weak field coordination environment consisting of O-atoms in the UiO-M (M = Co, Fe) system is very different than those accessible to molecular catalysts that typically consist of stronger field N- or P-donor atoms. Beyond the low coordination number, weak field catalyst sites exhibit high electrophilicity to lead to novel reactivities.
2. MOFs not only stabilize active Co-H species at SBUs via site-isolation, but also control the chemoselectivity. The steric hindrance from the three phenyl rings surrounding the Co center in the MOF directs selective binding of less hindered benzylic C-H bonds with Co-center in borylation and silylation reactions.
3. UiO-Co is much more active and chemoselective than homogeneous cluster control, $Zr_6(OCoCl)_4O_4(OMc)_{12}$, due to the beneficial effect of active site isolation in MOFs.
4. As robust and solid catalysts, UiO-Co catalyst can be recycled and reused multiple times. In addition, MOF-catalyst can be removed from organic products via simple filtration, affording pure products with very low metal contamination.

Despite the fact that the authors argue against this criticism their reply remains somewhat superficial. How can the similarity of PXRDs be a proof that BuLi does not react with a carboxylate? Certainly there are other more convincing experiments that can be carried out.

Response: We agree with the reviewer's point that the similarity of PXRDs cannot prove the stability of carboxylate linkers toward nBuLi. We have provided two additional experimental data to show that nBuLi did not react with the dicarboxylate bridging ligands of the MOFs.

1. NMR: To confirm the identity of ligand in the nBuLi-treated MOF (in the same manner as the UiO lithiation reactions), we digested the lithiated MOF with deuterated sulfuric acid and acquired 1H NMR spectrum of the solution. The 1H NMR spectrum revealed the presence of only terphenyldicarboxylate (TPDC) ligand and no sign of other aromatic products resulting from nucleophilic attack(s) on the carboxylate groups (Please see Supplementary Figure 8 and Supplementary Discussion 1 for experimental details).
2. IR: IR spectra of UiO-68-CoCl shows the presence of two strong carboxylate carbonyl stretching frequencies at 1605 and 1591 cm^{-1} similar to UiO-68 (Supplementary Figure 9a), indicating carboxylate groups remained intact after lithiation and metalation with $CoCl_2$.

We believe that, due to the more anionic nature of carboxylates of the $Zr_2(\mu_2-OOC)_2$ moiety, which makes the carbonyl carbons less electrophilic, and the structural rigidity of the MOF, the carboxylates are not prone to nucleophilic attack by nBuLi."

It should be avoided to draw too many pictures that are not supported by hard facts. Amorphous side products are not detected in PXRDs and thus we only see an excerpt of reality. Along the same line, statements arguing that BET surface areas are not important for catalysis are not very helpful

and not convincing. Given the slow reaction rates, minority species in solution redepositing on amorphous supports could be the major active species without a hint for detectability.

Response: We have performed several catalytic and control reactions to rule out any soluble or amorphous minor species as the active catalyst.

1. The reaction times can be drastically reduced by increasing the Co-loadings. Please see our response to the first comment.
2. The leaching of Co and Zr into the supernatant was very low for each type of reactions, which means the amount of leached species is insignificant and the leached species cannot account for the high activities exhibited by the UiO-Co catalysts. For example, in case of C-H borylation of p-xylene, the leaching of Co and Zr into the supernatant was only 0.14% and 0.056%, respectively, as shown by ICP-MS analysis.
3. No further conversion was detected after removal of UiO-Co from the reaction mixture (Supplementary Discussion 2 and 5). Moreover, borylation reaction did not occur even after the addition of UiO-68 to the supernatant, which rules out the possibility of any minor species in solution that might have redeposited on solid support as the active catalyst.
4. Our control experiment ruled out the contribution of catalytic activity of any trapped Co nanoparticles within the MOFs (Supplementary Discussion 3). In addition, nBuLi, UiO-68-MOF, lithiated UiO-68, and also Co-nanoparticles were all inactive in catalyzing benzylic C-H borylation reactions.

In conclusion, the correspondence raises more questions and is not helpful to present a clear picture of achievements.

Response: With the additional data provided, we believe that we have proved beyond any reasonable doubt about our main conclusion "The straightforward metalation of secondary building units (SBUs) of UiO-MOFs with readily available cobalt and iron precursors affords highly active and reusable single-site solid catalysts for a range of organic reactions, including chemoselective borylation, silylation, and amination of benzylic C–H bonds, as well as hydrogenation and hydroboration of alkenes and ketones."

Reviewers' comments:

Reviewer #2 (Remarks to the Author):

I am still not highly enthusiastic for this contribution due to the many open questions which are only partially clarified. The authors have now commented and added new experiments and probably done the best they can do to clarify the questions raised. In particular the huge drop in surface area is still a miracle and the indicated "lattice distortion" cannot account for that. The broadening of the Peaks indicates indeed some Degradation of the Framework structure, despite the fact that the origin of this cannot be explained. Some indication can be gained from the SEM data, since the Images, if analyzed in Detail, indicate a higher metal concentration of Co/Fe on the outer surface of the MOF particles. Such pore blocking or covering of the outer surface could indeed explain the decrease of surface area.

Regarding the catalytic results it seems borylation with 0,02 mol % catalyst have been reported in literature with TON = 5000.

In summary I would not give this paper the highest priority for publication in Nat. Commun.

Author response to reviewer comments

Reviewer #2 (Remarks to the Author):

I am still not highly enthusiastic for this contribution due to the many open questions which are only partially clarified. The authors have now commented and added new experiments and probably done the best they can do to clarify the questions raised. In particular the huge drop in surface area is still a miracle and the indicated "lattice distortion" cannot account for that. The broadening of the Peaks indicates indeed some Degradation of the Framework structure, despite the fact that the origin of this cannot be explained. Some indication can be gained from the SEM data, since the Images, if analyzed in Detail, indicate a higher metal concentration of Co/Fe on the outer surface of the MOF particles. Such pore blocking or covering of the outer surface could indeed explain the decrease of surface area.

Regarding the catalytic results it seems borylation with 0.02 mol % catalyst have been reported in literature with TON = 5000.

In summary I would not give this paper the highest priority for publication in Nat. Commun.

Response: We are thankful for the reviewer's critical comments, but we do not believe that there are many open questions in this manuscript as stated by the reviewer. We have addressed all of the concerns raised by this reviewer with hard experimental data in the last round of review, except the BET data. We would like to address the reviewer's additional comments below:

- 1) We have now carefully carried out nitrogen adsorption experiments in the past few weeks and observed significantly higher BET surface areas for both UiO-CoCl [1815 m²/g (new) vs. 1125 m²/g (old)] and UiO-FeBr [1160 m²/g (new) vs. 708 m²/g (old)]. We also realized that we made a mistake in measuring the surface area of UiO samples with my colleague's surface area analyzer. We re-measured the sorption data for the UiO on our own

instrument with the same settings as UiO-CoCl and UiO-FeBr, and obtained a surface area of 2815 m²/g (instead of 3318 m²/g for the old measurement). The differences between UiO and metalated UiO samples can be rationalized as follows.

UiO-CoCl: UiO-CoCl has an increased molecular weight after metalation (by 15%) and reduced pore sizes due to the presence of CoCl species (by ~15%). The expected surface area for UiO-CoCl would be $\sim 2815/(1.15 \times 1.15) = 2129$ m²/g, which is lightly larger than the experimental value of 1815 m²/g. The minor discrepancy of ~14.7% is likely due to distortion of the framework upon removal of the solvent, as suggested by the broadening of the PXRD pattern of UiO-CoCl after BET analysis.

UiO-FeBr: UiO-FeBr has an increased molecular weight after metalation (by 21%) and reduced pore sizes due to the presence of CoCl species (by ~21%). The expected surface area for UiO-FeBr would be $\sim 2815/(1.21 \times 1.21) = 1923$ m²/g, which is lightly larger than the experimental value of 1160 m²/g. The discrepancy of ~39.7% is likely due to distortion of the framework upon removal of the solvent, as suggested by the broadening of the PXRD pattern of UiO-CoCl after BET analysis.

- 2) We respectfully disagree with the reviewer's comment "The broadening of the Peaks indicates indeed some Degradation of the Framework structure." In fact, in a *J. Am. Chem. Soc.* article (2011, 133, 18257), Matzger and coworkers showed that a densified surface layer of MOF-5 prevented the entry of even small molecular species into the crystal framework, leading to reduced surface area probed by N₂ adsorption. We showed in our *Nat. Chem.* article (2010, 2, 838) that N₂-sorption based surface areas have little value in describing large pore/channel MOFs in liquid phase catalysis. We do not dry our MOF samples for liquid-phase catalysis reported in this paper. The distortion seen for the BET sample would not have happened for the MOF samples used for catalytic reactions.
- 3) We thank the reviewer for critically examining our TEM/EDX data. However, we respectfully disagree with the reviewer's interpretation; we know that the MOF particles have irregular morphologies with different thicknesses across the particles. In this case, we do not expect the EDX mapping to show uniform intensities at the center and along the edge. The only information we can extract from TEM/EDX is that Co and Fe appear to be evenly distributed through the MOFs.
- 4) The reviewer's statement "Regarding the catalytic results it seems borylation with 0.02 mol % catalyst have been reported in literature with TON = 5000" in fact incorrect. The paper cited by the reviewer (*J. Am. Chem. Soc.*, 2014, 136, 4133) reported arene C-H (sp²) borylation with homogeneous Co catalysts. Our paper reports much more difficult benzylic (sp³) C-H borylation, which was unprecedented when we originally submitted our paper. During the revision of our manuscript, one paper appeared in *J. Am. Chem. Soc.* (2016, 138, 766) reporting uncontrolled mono-, di-, and tri-borylation of benzylic C-H bonds with 5-50 mol% homogeneous Co catalyst.

In summary, we stand by our belief that was conveyed in our original cover letter: The simplicity and efficacy of this novel M@SBU strategy can inspire the synthesis of other earth-abundant metal catalysts for sustainable chemical catalysis. The cost-effective nature of the present approach can potentially lead to practical MOF catalysts, paving the way for translating MOFs from novel discoveries to much needed industrial applications.